# Isolation, Screening, and Identification of Alkaline Protease-Producing Bacteria and Application of the Most Potent Enzyme from *Bacillus* sp. Mar64

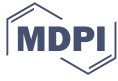

Essam Kotb [1,2,*], Amira H. Alabdalall [1,2], Mariam A. Alsayed [1,2], Azzah I. Alghamdi [1,2], Eida Alkhaldi [1,2], Sayed AbdulAzeez [3] and J. Francis Borgio [3,4]

1    Basic and Applied Scientific Research Center (BASRC), Imam Abdulrahman Bin Faisal University (IAU), P.O. Box 1982, Dammam 31441, Saudi Arabia; aalabdalall@iau.edu.sa (A.H.A.); maaalsayed@iau.edu.sa (M.A.A.); azalghamdi@iau.edu.sa (A.I.A.); ealkhaldi@iau.edu.sa (E.A.)
2    Department of Biology, College of Science, Imam Abdulrahman Bin Faisal University (IAU), P.O. Box 1982, Dammam 31441, Saudi Arabia
3    Department of Genetic Research, Institute for Research and Medical Consultations (IRMC), Imam Abdulrahman Bin Faisal University (IAU), P.O. Box 1982, Dammam 31441, Saudi Arabia; asayed@iau.edu.sa (S.A.); fbalexander@iau.edu.sa (J.F.B.)
4    Department of Epidemic Diseases Research, Institute for Research and Medical Consultations (IRMC), Imam Abdulrahman Bin Faisal University (IAU), P.O. Box 1982, Dammam 31441, Saudi Arabia
*    Correspondence: ekghareeb@iau.edu.sa or ekotb@hotmail.com

**Abstract:** In this study, thirty-seven alkaline protease-producing bacteria were recovered from different regions of Saudi Arabia. The proteolytic strain with the highest productivity was identified as *Bacillus* sp. Mar64. Maximum productivity of Mar64P alkaline protease was reached at 60 h, pH 9.0, and 45 °C using 1% tyrosine and 0.5% maltose as nitrogen and carbon supplies, respectively. Specific activity was intensified to 8.5-fold with a recovery of 12.4% and SDS—PAGE revealed one band at 28 kDa after enzyme purification. Mar64P was maximally active at 55 °C and pH 11.0 with thermal stability up to 70 °C and pH stability at 7.0–12.0 for 1 h. It was inhibited by EDTA and unaffected by PMSF, therefore tentatively classified as metalloprotease-type. Storage efficacy was effective for up to eight weeks and it was durable in presence of organic solvents (20%, *v/v*) such as acetonitrile, acetone, and isopropanol upto to 15 days. The enzyme was compatible with dry detergents at both low and high temperature, in addition, was successful in removing various stains such as blood, egg yolk, chocolate, tea, coffee, and sweat. Furthermore, it was successful in removing skin hairs and hydrolyzing gelatin of waste X-ray films. Collectively, due to these unique properties, Mar64P could be considered an environmentally friendly candidate in both detergent and leather industries.

**Keywords:** alkaline protease; destaining; detergent compatibility; protein wastes



## 1. Introduction

Proteolytic enzymes trigger the hydrolysis of proteinaceous compounds and wastes into shorter peptides or even free amino acid residues [1]. They are classified according to the location of cleavage into exoproteases and endoproteases. When the cleavage occurs near the terminals, they are termed exoproteases, while when it occurs as far as five residues from the terminals they are termed endoproteases. They are also classified according to their functional groups at the active sites into threonine proteases (E.C. 3.4.25), metalloproteases (E.C. 3.4.24), aspartic proteases (E.C. 3.4.23), cysteine proteases (E.C. 3.4.22), and serine proteases (E.C. 3.4.21). According to the optimum reaction pHs, they are subdivided into acidic proteases, neutral proteases, and alkaline proteases [2].

Proteases are discovered in all living organisms, including eukaryotes such as animals, plants, fungi, protists, and the prokaryotic domains bacteria and archaea, where they perform essential functions in their metabolism. In addition, they are very valuable

commercially representing sixty percent of total enzyme sales worldwide, and are expected to gain about fifteen billion US dollars per year by 2025. The sales of alkaline proteases alone represent thirty-five percent of all commercial proteases [3]. Bacteria and fungi are the favored supply for alkaline proteases and other commercial enzymes due to the ease of their cultivation, maintenance, genetic transformation, and are not affected by seasonal variations. In addition, they can be produced on a large scale using cheap culture media involving agro-industrial wastes and produced outside the cells as extracellular metabolites facilitating the downstream separation processes [4].

The reason for their widespread applications in pharmaceutical, detergent, food, leather, silk, and cosmetic industries resides in their activity and stability at elevated pHs of 9–12 [5]. They are eco-friendly alternatives for many harmful chemicals used in detergent formulations [1]. They become an essential ingredient in almost all detergent formulations to improve washing performance by removing protein stains from blood, milk, and other foods. Their activity and stability in detergent formulations depend on several considerations including detergent composition, presence of inhibitors, presence of oxidants, washing temperature, and pH [6].

Most alkaline proteases are produced from genus *Bacillus* such as *B. licheniformis* and *B. subtilis*. However, many of them have low stability in heterogeneous environmental conditions. However, the multiplicity and variations of microbial flora populating the earth make it feasible to recover new microorganisms producing alkaline proteases with the desired commercial characteristics [7]. There aren't enough screening programs for alkaline protease-producing bacteria in the Gulf region and there are no published reports on the application of alkaline proteases from *Bacillus* sp. Mar64 in industrial applications.

Given the prominence on converting industrial activities environmentally sustainable, the current research is directed to isolate bacteria from the local environment of Saudi Arabia capable of producing alkaline proteases under extreme conditions. In addition, the applications of alkaline protease produced by a recently recovered bacterium *Bacillus* sp. Mar64 as a detergent additive, hydrolyzer of waste X-ray films, and dehairing agent of animal skins were assessed.

## 2. Materials and Methods

### 2.1. Materials

Sephadex G-100 FF, DEAE-Sepharose CL-6B, and SDS—PAGE chemicals were obtained from Pharmacia Biotech (Uppsala, Sweden). Azocasein, EDTA, and PMSF were bought from Sigma—Aldrich (St. Louis, MO, USA). Other compounds were of analytical grade and were purchased from some local providers.

### 2.2. Sampling and Isolation Procedure

From January to March 2019, thirty soil samples were collected from several regions of Saudi Arabia including Khobar, Dammam, Anak, Jubail, Ras Tanura, Qatif, and Saihat from the eastern region, Riyadh from the central region, and Taif and Makkah from the western region. The decimal serial dilution method was accomplished to reduce the bacterial concentration. To select alkaline protease producers, skim milk agar consisting of (g/L) agar (20), yeast extract (1), peptone (5), beef extract (1), and NaCl (5) was used. Exactly 800 mL of distilled water was added and after sterilization, 200 mL of sterile skim milk was added. The initial pH was corrected to 9.0 and incubation was done at 40 °C for 24 h. By the end of the incubation period, individual colonies showing cleared halos were picked for purification by the quadrate streaking method. The purified cultures were maintained on nutrient agar at 4 °C, and into 20% (*v/v*) glycerol at −80 °C for short-term and long-term preservation, respectively.

## 2.3. Enzyme Production and Assay

Inocula of the highest promising isolates were allowed to grow overnight in Luria-Bertani (LB) broth consisting of (g/L) NaCl (5), yeast extract (5), and tryptone (10) with pH equilibrated at 7.0. Exactly, 2% (*v/v*) inoculum ($1.0 \times 10^8$ cfu/mL) of overnight old cultures grown in LB broth was utilized to inoculate 250 mL flasks holding 50 mL of basal medium consisting of (g/L) casein (10), glucose (5), yeast extract (0.4), NaCl (3), $K_2HPO_4$, (8.19), $KH_2PO_4$ (0.41), $MgSO_4.7H_2O$ (0.5), $CaCl_2$ (0.1), $MnSO_4$ (0.1), $FeSO_4$ (0.02), and $ZnSO_4$ (0.02). The initial pH was altered to 9.0 and the flasks were incubated for 48 h at 40 °C with shaking at 120 rpm. By the end of incubation, crude enzymes were prepared by centrifugation at $2000\times g$ for 20 min in a cooling centrifuge (Hermle Labortechnik GmbH, Z 326 K, Wehingen, Germany).

The protein content was measured directly at $A_{280}$ nm and the proteolytic activity was measured against the substrate azocasein. Five-hundred microliter of enzyme preparation was mixed with 500 μL 0.8% (*w/v*) azocasein solution of pH 9.5 and incubated at 40°C for 30 min. Two milliliters of 10% (*w/v*) trichloroacetic acid (TCA) were included to terminate the reaction and then incubated for 30 min in an ice bath. For the separation of unreacted substrate molecules, centrifugation was done at $2000\times g$ for 20 min. For quantification of the reaction product, 1 mL of supernatant was associated with 1 mL of 1 M NaOH then optical density was assessed at $A_{440}$ nm wavelength (Spectrophotometer UV5Bio, Mettler Toledo, Stockholm, Sweden). Blank was prepared with the same procedures, with enzyme replaced by the same volume of 0.05 M glycine-NaOH buffer (pH 9.5). The unit of alkaline protease was quantified by subtracting the value sample absorbance from the reading of the blank and each 0.01 optical density was represented as one unit of enzyme activity [8].

## 2.4. Molecular Characterization of the Most Potent Bacterial Isolates

DNA was extracted from target bacteria and the *16SrDNA* gene was amplified using the forward and reverse primers (F: 5′-AGAGTTTGATCCTGGCTCAG-3′; R: 5′-TACGGC TACCTTGTTACGACTT-3′) and was carried out with PCR master mix (MOLEQULE-ON, Auckland, New Zealand) using Biometra T-Professional thermocycler (Biometra, Goettingen, Germany) with an annealing temperature of 56 °C for 35 cycles. PCR amplicons were purified using QIAquick PCR Purification Kit (Qiagen, Hilden, Germany) and were sequenced with the same forward and reverse primers separately using BigDye® Terminator v3.1 Cycle Sequencing Kit and 3500 genetic analyzers, (Applied Biosystems, Foster City, CA, USA). The *16SrDNA* gene sequence of forward and reverse sequences were merged, submitted to GenBank and analyzed for the identification of the bacterial species using the EzBioCloud server and the phylogenetic tree was constructed UPGMA method and the Maximum Likelihood method using MEGA11 software as described earlier [9]. The species identification as *Bacillus* sp. was completed based on the metabolic fingerprint of the bacterium.

## 2.5. Optimization of Production Conditions

Based on the initial productivity of enzyme during this survey, isolate Mar64 was selected for completion of this study. Parameters such as incubation period, fermentation pH, fermentation temperature, carbon source, and nitrogen source in the medium were adjusted to achieve maximal alkaline protease productivity. Each factor was studied separately, the highest productivity was measured, and the optimum result was determined to move to the next parameter. Protease productivity was determined at $A_{440}$ nm and protein quantification was determined at $A_{280}$ nm.

## 2.6. Enzyme Purification

Bacterial cells of the most potent isolate were separated from the crude enzyme present in 1000 mL culture broth by centrifugation at $2000\times g$ for 15 min. The alkaline protease designated as MarP64 was then precipitated by salting out with 60% ammonium sulphate followed by anion-exchange and gel filtration chromatography through DEAE-Sepharose

CL-6B and Sephadex G100 FF columns, respectively [10]. Protein pellets due to ammonium sulphate treatment were harvested after 24 h incubation at 4 °C by cooling centrifuge at $7000 \times g$ for 15 min. The precipitate was vortexed with 6 mL of 0.2 M sodium phosphate buffer (pH 7.4, buffer A). The included ammonium sulphate and other salts were removed from proteins by dialysis against the same buffer. The intense dialysate was added onto a DEAE-Sepharose CL-6B gel (column dimension of $0.5 \times 5$ cm$^2$) balanced with the same buffer solution. Elution was done with 0.2 M sodium borate buffer (pH 9.4, buffer B) with a linear rise of 0–1 M NaCl solution. Fractions showing alkaline protease activity were pooled and applied to the next column packed with Sephadex G100 FF (column dimension of $2.0 \times 45$ cm$^2$). The elution frequency was adjusted at 0.5 mL/min. Active fractions were mixed and lyophilized for further characterization. The molecular weight and purity of MarP64 were determined by SDS—PAGE involving 15% (*w/v*) separating and gel 5% (*w/v*) stacking gel [11].

### 2.7. Biochemical Properties of Enzyme

### 2.7.1. Effect of Temperature on Enzyme Activity and Stability

The best temperature for enzyme catalysis was assessed by incubating the reaction mixtures at 20–70 °C for 30 min. Thermal stability was assessed by incubating the enzyme preparations without substrate at 50–90 °C for 1 h. By the end of incubation, residual activity against the substrate was evaluated at 55 °C for 30 min. Proteolytic activity was assayed as mentioned in the assay subsection. The enzyme that wasn't incubated served as a blank for this test.

Residual activity = activity of the sample $\times$ 100] $\div$ activity of blank [10].
Relative activity = [activity of the sample $\times$ 100] $\div$ maximum activity.

### 2.7.2. Effect of pH on Enzyme Activity and Stability

Alkaline protease activity was tested at pH 5–12 using an appropriate buffer solution by incubating the reacting mixtures at the standard settings for 30 min at 55 °C. Finally, the enzymatic activity was verified. To achieve the full range of experimentation, 100 mM acetate buffer was used at pH 5–6, 100 mM sodium phosphate buffer for pH 6–8, 100 mM glycine-NaOH buffer for pH 8–11, and 100 mM hydroxide-chloride buffer for pH 11–12. For the pH stability test, preincubation of buffered enzyme fractions lacking substrate was done at 55 °C for 1 h at pH 5.0–12.0, then remaining activity was evaluated at the optimized conditions.

### 2.7.3. Effect of Metal Ions and Protease Inhibitors

Metal ions such as $Ca^{2+}$, $Mn^{2+}$, $Fe^{2+}$, $K^+$, $Mg^{2+}$, and $Zn^{2+}$ were tested by incubation of Mar64P preparations with metals at 0.05% (*w/v*) concentration for 1 h at 55 °C. In addition, the effect of protease inhibitors phenyl methyl sulphonyl fluoride (PMSF) and ethylene diamine tetra acetic acid (EDTA) were studied at 5 mM. A blank was carried out without metal ions or inhibitors. The assay method was different from the previous due to the color change of assay solutions triggered by the metal ions. Therefore, the well plate method was adopted. Exactly, 100 µL of enzyme is loaded into a 10 mm diameter well of skim milk plates after preparing and sterilizing the medium. The unit (**U**) of protease activity was quantified as the quantity of protease required to create 1.0 mm clear area around the well. The effect of EDTA concentration (1.0 to 40.0 mM) on the enzymatic stability was studied by incubation with Mar64 preparations at 55 °C for 1 h. The remaining activity was assessed by addition of the substrate azocasein and following the standard settings of reaction.

### 2.7.4. Effect of Storage Conditions and Organic Solvents

This experimentation was performed to assess the shelf-life of tested enzyme under storage temperatures. Enzyme fractions were stored at 4 °C and 25 °C for eight weeks without any additives and residual activity was determined weekly [12]. In addition, the impact of organic solvents such as acetonitrile, acetone, ethanol, n-hexane, toluene, and

isopropanol on the stability of enzyme was assessed. Aliquots of 4.0 mL of enzyme solution were mixed with 1.0 mL of organic solvent (20%, *v/v*) and were incubated at 4 °C and 25 °C for 15 days with shaking speed of 120 rpm, and the residual activity was evaluated at the end of incubation period [13].

### 2.8. Detergent Compatibility

The constituent enzymes involved in powdered detergents (Persil ProClean®, Dac Extra®, Bonux 3-in-1®, Omo Laundry Detergent®, Tide Simply Clean & Fresh®, and Ariel Matic Front Load Washing®) were denatured at the beginning of the experiment by autoclaving for 15 min at 121 °C. The alkaline protease preparation (150 U enzyme/mL) was homogenously mixed with the powdered detergents (7 mg detergent/mL) and incubated for 1 h at temperatures 25 °C and 55 °C. The remaining enzymatic activity was assessed, and the relative activity was determined by considering an equivalent volume of distilled water mixed with enzyme as a blank (100% activity) [14].

### 2.9. Destaining of Cotton Fabrics

White cotton fabrics of ~3 × 3 cm$^2$ dimension were stained by ~100 μL of blood, eggs, chocolate, tea, coffee, and sweat. After the dryness of stains at 60 °C for 60 min, they were immersed in a solution of one set of the experiments at pH 11.0. The sets were (1) distilled water as a control, (2) detergent solution at 7 mg/mL concentration, and (3) denatured detergent solution plus enzyme preparation at a concentration of 150 U/mL. Incubation was allowed at room temperature (25 °C) for 1 h with a shaking speed of 120 rpm. By the end of incubation, fabrics were air-dried after being rinsed with tap water, and the stain elimination effectiveness was determined by visual inspection of the color intensity of cloth [15].

### 2.10. Dehairing Activity

A skin piece of sheep was obtained from a local slaughterhouse then washed with distilled water to eliminate impurities and dried for 30 min at 50 °C. It was cut into smaller pieces of ~4 × 4 cm$^2$. Pieces were then immersed in 150 U Mar64P/mL at pH 11.0 and 55 °C for 3 h with a shaking speed of 120 rpm. Dehairing activity was checked at time intervals of 30 min by shedding the hair using sterile forceps [16].

### 2.11. Hydrolysis of Gelatin and Recovery of Silver from Waste X-ray Films

A waste X-ray film was cleaned with distilled water, then wiped with a cotton swab soaked in ethanol and dried for 2 h at 40 °C. The sheet was sliced into ~4 × 4 cm$^2$ pieces of equal sizes. Pieces were then immersed in flasks containing 150 U Mar64P/mL at pH 11.0 and incubation was carried out at 55 °C for 3 h with a shaking speed of 120 rpm. Films incubated in 100 mM sodium hydrogen phosphate-NaOH Buffer (pH 11) served as control. The progress of gelatin hydrolysis during the exposure period was assessed by visual inspection of the detachment of the gelatin layer and the appearance of turbidity due to the release of silver atoms [17].

### 2.12. Statistical Analysis

Unless otherwise declared, all tests were carried out in three replications, statistically evaluated by Microsoft Excel Spreadsheet Software, Microsoft 365, and the results have been presented as means ± SD.

## 3. Results

### 3.1. Isolation of Alkaline Protease-Producing Bacteria

In this research, one hundred-thirty isolates were recovered from soil samples collected from many places in Saudi Arabia. Among them, thirty-seven alkaline protease-producing bacteria were selected after incubation at 40 °C for 24 h in skim milk agar of pH 9.0. The most potent isolates were numbers 64, 9, 67, and 77, respectively while the lowest producers were numbers 95 and 107. Interestingly, isolates 9, 64, and 67 were from soil samples obtained from two farms at Saihat, while isolate 77 was recovered from a soil sample from a farm at Dammam (Table 1). See the Supplementary Material for the geographic localites of soil samples and protease-producing bacteria (Supplementary Figures S1 and S2 in addition to Supplementary Tables S1 and S2).

**Table 1.** Most potent alkaline protease-producing bacterial isolates in this study. The assay of enzyme productivity was based on the spectrophotometric measurements at $A_{440}$ using 0.8% (*w/v*) azocasein as a substrate.

| Isolate Number | Source Locality | Alkaline Protease Productivity (U/mL) | Strain Characterization | GenBank Accession Number |
|---|---|---|---|---|
| 9 | Saihat farm 1 | 142.8 ± 16.2 | *Enterobacter* sp. Mar9 | OP429622.1 |
| 64 | Saihat farm 2 | 159.9 ± 7.2 | *Bacillus* sp. Mar64 | OP429623.1 |
| 67 | Saihat farm 2 | 137.6 ± 7.2 | *Bacillus* sp. Mar67 | OP429624.1 |
| 77 | Dammam farm | 92.6 ± 7.6 | *Bacillus* sp. Mar77 | OP429625.1 |

Gram's staining showed that isolates Mar64, Mar67, and Mar77 were Gram-positive rods while isolate Mar9 was a Gram-negative rod. When chromosomal DNA was extracted the *16SrDNA* gene was amplified and electrophoresed on 1% agarose gel and confirmed for the absence of nonspecific amplicons. Sequencing of the purified amplicon was performed in both directions and merged to trim the repeating sequences. The final definition of the four bacterial isolates becomes *Enterobacter* sp. Mar9 (accession number OP429622.1), *Bacillus* sp. Mar64 (accession number OP429623.1), *Bacillus* sp. Mar67 (accession number OP429624.1), and *Bacillus* sp. Mar77 (accession number OP429625.1). GenBank accessions were presented in the form of phylogenetic trees using the EzBioCloud. The phylogenetic tree of isolate Mar64 is presented in Figure 1. Estimates of Evolutionary Divergence between Sequences of strain *Bacillus* sp. Mar64 using *16SrDNA* gene and reference sequences using the Maximum Composite Likelihood model through MEGA11 is shown in Supplementary Table S3. The metabolic fingerprint of the most potent isolate Mar64 was as follow: It was positive to casein hydrolysis, citrate utilization, indole, $H_2S$, and acetoin production. In addition, it was positive to enzymes such as catalase, cytochrome oxidase, and gelatinase while negative to urease, tryptophan deaminase, ornithine decarboxylase, and lysine decarboxylase. It was able to produce acid from arbutin and cellobiose while no acid produced from amygdalin, arabinose, and arabitol.

### 3.2. Optimization of Alkaline Protease Productivity

Table 2 shows the optimized conditions for achieving the highest alkaline protease productivity by *Bacillus* sp. Mar64 when allowed to grow on the basal medium. By the end of this step, enzyme productivity reached 172.4 U/mL which represented a 2.78-fold increase compared with the initial production conditions (61.9 U/mL).

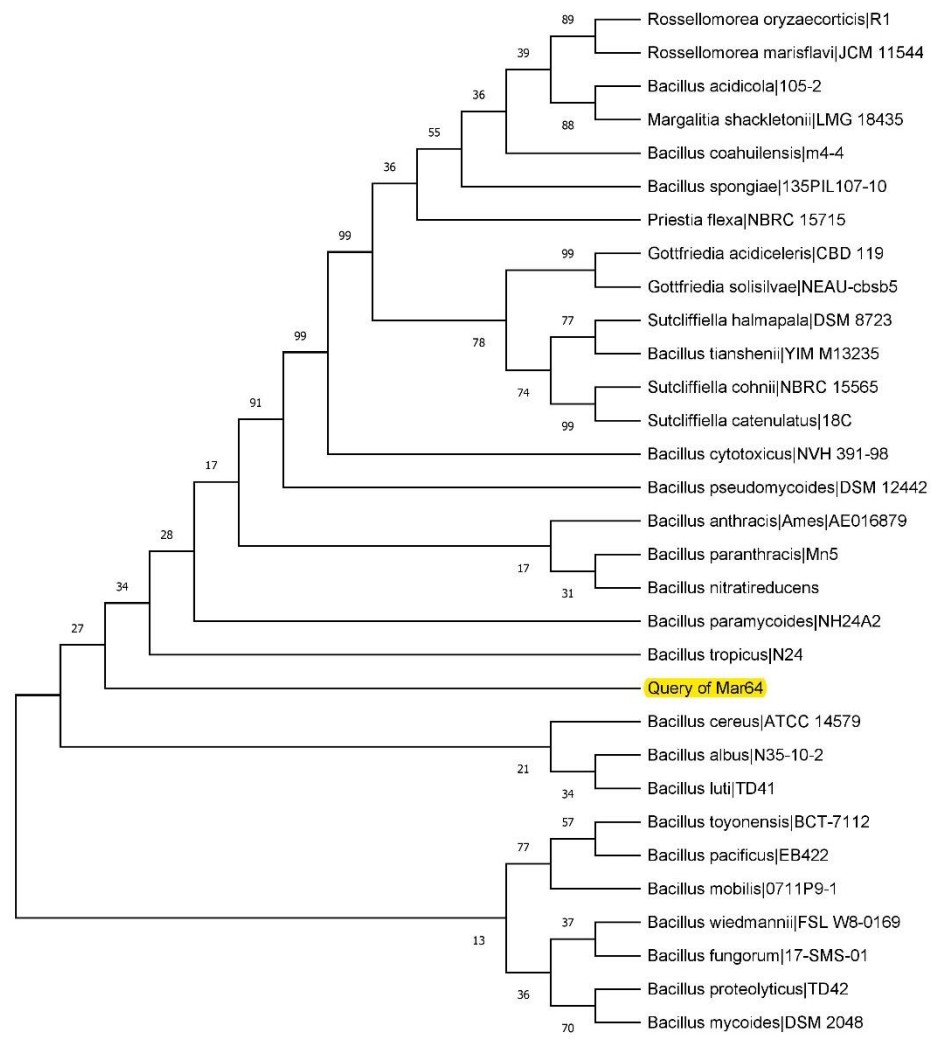

**Figure 1.** Phylogenetic tree of strain *Bacillus* sp. Mar64 using *16SrDNA* gene (Accession number OP429623.1).

**Table 2.** Summary of the optimized conditions for maximal alkaline protease productivity by *Bacillus* sp. Mar64. The initial production was done in a basal medium consisting of (g/L) casein (10), glucose (5), yeast extract (0.4), NaCl (3), $K_2HPO_4$, (8.19), $KH_2PO_4$ (0.41), $MgSO_4.7H_2O$ (0.5), $CaCl_2$ (0.1), $MnSO_4$ (0.1), $FeSO_4$ (0.02), and $ZnSO_4$ (0.02) with pH adjusted at 9.0 and the incubation was done at 40 °C for 48 h.

| Parameter | Optimal Value | Maximal Productivity (U/mL) | Fold (×) |
|---|---|---|---|
| Unoptimized conditions | NA | $61.9 \pm 2.2$ | 1.0 |
| Incubation period | 60 h | $93.2 \pm 3.0$ | 1.5 |
| Initial pH | 9.0 | $110.1 \pm 7.0$ | 1.8 |
| Fermentation temperature | 45 °C | $155.3 \pm 14.4$ | 2.5 |
| Carbon source | 0.5% maltose | $162.1 \pm 8.4$ | 2.6 |
| Nitrogen source | 1% tyrosine | $172.4 \pm 18.0$ | 2.8 |

### 3.3. Enzyme Purification

Mar64P was separated and purified by consecutive purification steps mentioned in Table 3. The ending specific activity of protease became 8.5-fold with a recovery of 12.4% as judged by the original strength of enzyme in the cell-free supernatant of *Bacillus* sp. Mar64. SDS—PAGE analysis of the last fractions confirmed a prominent band at 28 kDa (Figure 2).

**Table 3.** Summary of purification steps.

| Purification Step | Total Activity (U) | Specific Activity (U/mg) | Recovery (%) | Purification (Fold) |
|---|---|---|---|---|
| Cell-free supernatant | 172,413.2 | 90.6 | 100.0 | 1.0 |
| 60% ammonium sulphate precipitation | 90,145.1 | 190.2 | 52.3 | 2.1 |
| Anion exchanger (DEAE-Sepharose CL-6B) | 53,312.0 | 571.8 | 30.9 | 6.3 |
| Gel permeation (Sephadex G-100 FF) | 21,368.6 | 769.1 | 12.4 | 8.5 |

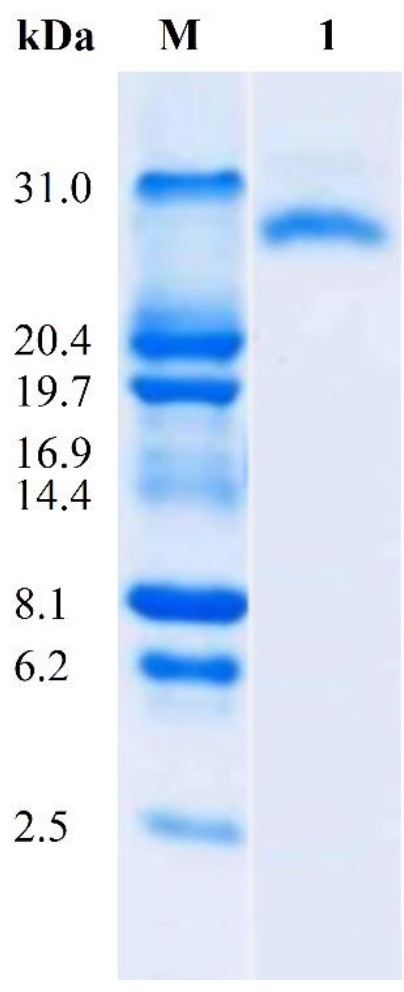

**Figure 2.** SDS—PAGE analysis of the purified Mar64P using 15% (*w/v*) separating gel and 5% (*w/v*) stacking gel. Lane M signifies the protein markers and Lane 1 corresponds to the purified Mar64P.

### 3.4. Biochemical Properties of Alkaline Protease

The tested protease showed highest activity at higher temperatures and was optimal at 55 °C (Figure 3a). In addition, it was thermally stable up to 70 °C for 1 h and drastically lost stability above 70 °C (Figure 3b). The results displayed in Figure 3c showed that best protease activity was at pH 11.0 and the enzyme was highly stable within a wide pH range of 7–12 for 1 h.

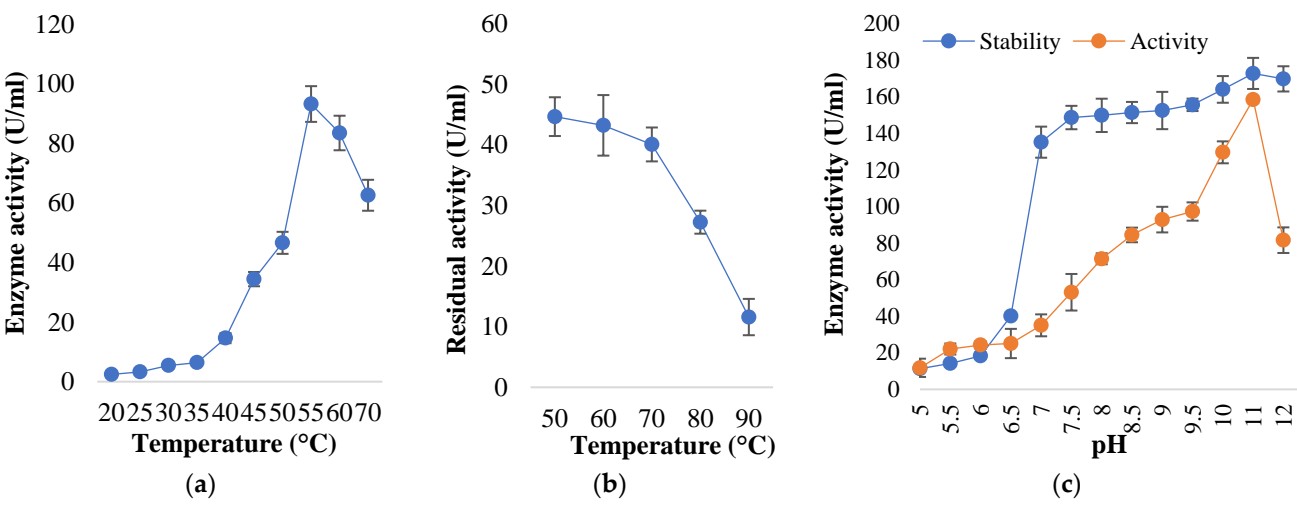

**Figure 3.** Impact of temperature on enzyme activity (**a**) and stability after 60 min exposure (**b**). Impact of pH value on protease activity and stability (**c**).

### 3.5. Effect of Metal Ions and Protease Inhibitors

All tested cations were stimulatory for the alkaline protease activity and was highest in case of 0.05% (*w/v*) $Mg^{2+}$ ions (Figure 4a). However, the proteolytic activity was suppressed by addition of 5 mM EDTA and unchanged with addition of 5 mM PMSF. The impact of EDTA concentrations on enzymatic activity was examined by incubation of enzyme preparations at 55 °C for 1 h with different concentrations (1.0 to 40.0 mM). The enzyme drastically lost its original activity by increasing EDTA concentration (Figure 4b).

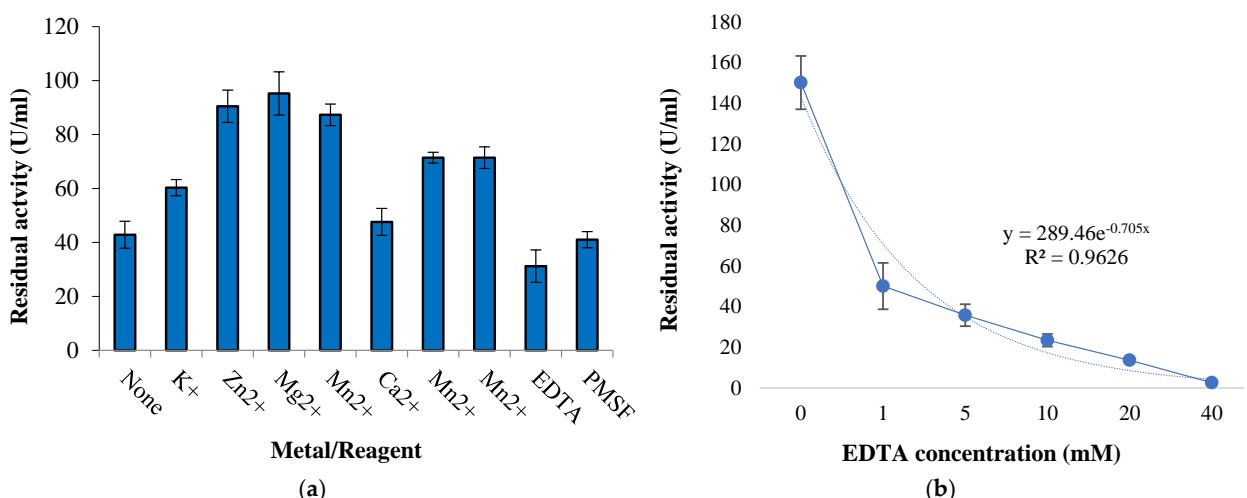

**Figure 4.** Impact of cations and protease inhibitors on Mar64 activity. (**a**) All cations were tested at 0.05% (*w/v*) concentration, while, the protease inhibitors EDTA and PMSF were tested at 5 mM concentration. The impact of EDTA concentration (1.0 to 40.0 mM) on enzymatic activity is displayed in panel (**b**) A blank was carried out without metal ions or inhibitors. Incubation was done for 1 h at 55 °C then remaining activity was assessed.

### 3.6. Storage Stability of Mar64P

Upon storage of enzyme preparations at 4 °C and 25 °C without additives, it was observed that protease remained stable with a slight decrease up to the 8th week at 4 °C (79.9 U/mL). While at 25 °C it retained stability up to the 4th week (83.5 U/mL) (Figure 5a). The stability of Mar64 protease was almost conserved after 15 days in the existence of 20% (*v/v*) organic solvent, especially in case of acetone, acetonitrile, and isopropanol. In the presence of toluene, the stability of Mar64P decreased drastically (Figure 5b).

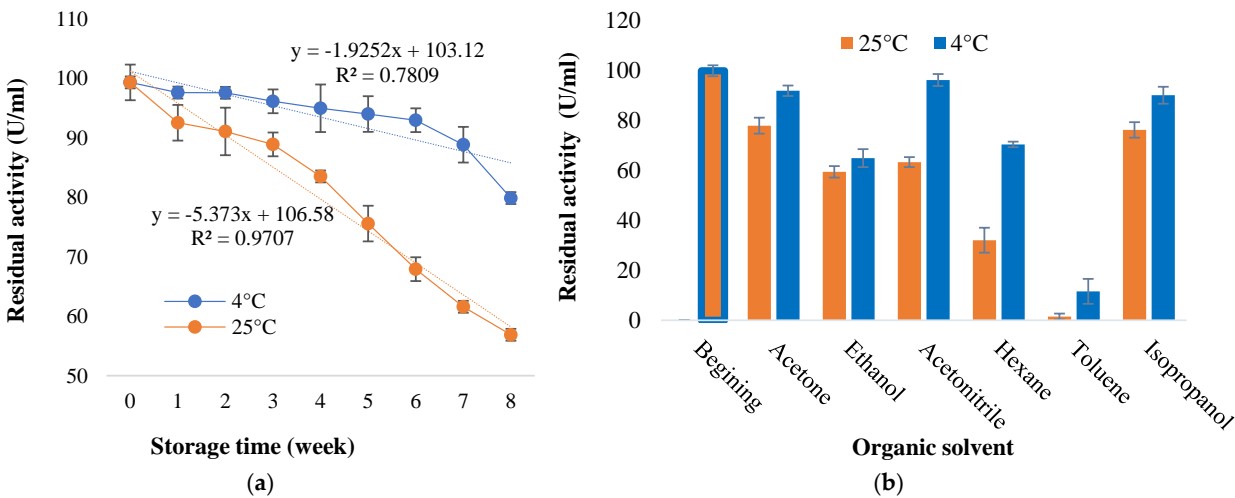

(**a**)                     (**b**)

**Figure 5.** Shelf-life of alkaline protease in the absence of additives (**a**) and in the existence of organic solvents (**b**) at 25 °C and 4 °C.

### 3.7. Commercial Application of Mar64P

#### 3.7.1. Detergent Compatibility

Dry detergents were tested with the alkaline protease at elevated temperature (55 °C) and room temperature (25 °C). At low temperature, the enzyme showed the highest compatibility with Omo™ and the lowest with Bonux™. At 55 °C the enzyme showed highest compatibility with Persil™ and Omo™ and lowest with Bonux™ (Figure 6).

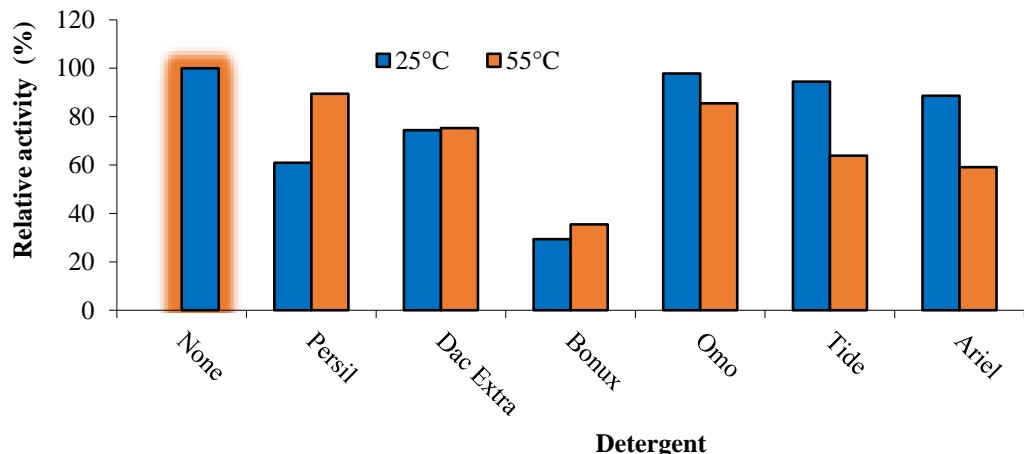

**Figure 6.** Compatibility of Mar64P with selected dry detergents at room temperature (25 °C) and high temperature (55 °C). The alkaline protease was mixed with the powdered detergents (7 mg detergent/mL) and incubated for 1 h. The relative activity was calculated by considering an equivalent volume of distilled water mixed with the enzyme as a blank.

### 3.7.2. Destaining of Cotton Fabrics

The ability of alkaline protease as an additive with the powdered detergent OMO™ to eradicate various contaminations from cotton fabrics stained with blood, egg, chocolate, coffee, tea, and sweat at temperature 25 °C was observed (Figure 7). Visual inspection of better cleaning showed that Mar64P plus detergent showed enhanced wash performance as compared with the detergent alone. Better cleaning was found in case of egg stains, coffee stains, and blood stains, while lowest enzymatic performance was observed in case of chocolate stains (Figure 7).

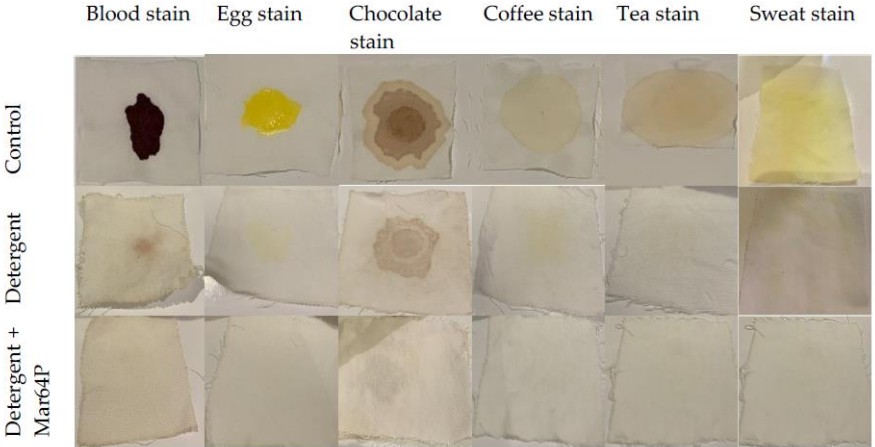

**Figure 7.** Wash performance of Mar64P against different stains on white cloth fabrics. Blood, eggs, chocolate, tea, coffee, and sweat stains were used. Each piece was immersed in a solution of one set of experiments at pH 11.0. The sets were (1) distilled water as a control, (2) detergent solution at 7 mg/mL concentration, and (3) detergent solution plus enzyme preparation at concentration 150 U/mL. Incubation was allowed at room temperature (25 °C) for 1 h.

### 3.7.3. Dehairing Activity

3 × 3 cm skin tissues were soaked in enzyme at 55 °C. Over time, the amount of detaching hair became higher. After 90 min, a saturation of skin tissue with the alkaline protease seemed to occur where the dehairing was highest. The external solution became turbid after 120 min due to the degradation of some skin proteins that loosened the connective tissue (Figure 8).

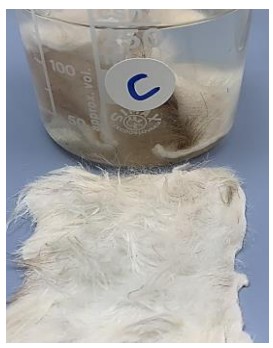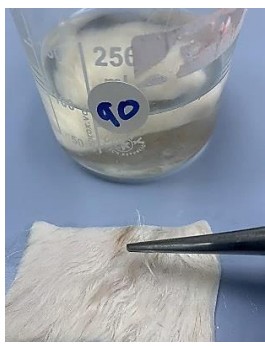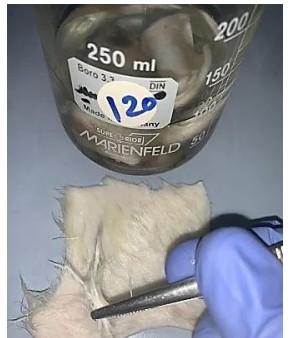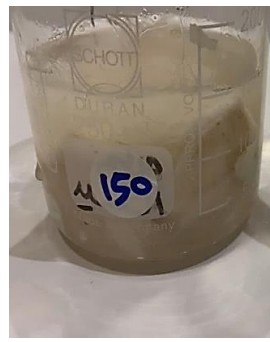

**Figure 8.** Dehairing activity of Mar64P against pieces of sheep's skin for 150 min exposure. Skin pieces were immersed in a solution of 150 U/mL of Mar64P at pH 11.0 and incubated at 55 °C for 3 h with a shaking speed of 120 rpm. Dehairing activity was checked at time intervals of 30 min by shedding the hair using sterile forceps. Control (C) represents the untreated skin piece with the enzyme. The remaining solution after 120 min exposure was turbid due to the beginning of degradation of skin proteins.

### 3.7.4. Degradation of Gelatin of Used X-ray Films

X-ray sheet pieces were immersed in 150 U alkaline protease /mL at pH 11.0 and 55 °C. It was observed that gelatin hydrolysis increased as time passes, and degradation was almost completed after 120 min of exposure (Figure 9).

**Figure 9.** Decomposition of the gelatin layer of used X-ray films. Pieces of X-ray films were immersed in flasks containing the 150 U/mL enzyme preparation at pH 11.0 and incubation was carried out at 55 °C for 3 h with a shaking speed of 120 rpm. With time the process of gradual removal of gelatin layer and the release of silver atoms by the alkaline protease were observed. The control test wasn't treated with enzyme.

## 4. Discussion

Screening for new alkaline protease-producing bacteria from unexplored localities may lead to novel producers with greatly active and stable enzymes that could be useful in many commercial applications. In this study, we have isolated thirty-seven alkaline protease-producers, the most potent among which was identified based on *16SrDNA* gene fingerprint as *Bacillus* sp. Mar64. The produced alkaline protease was designated as Mar64P. In fact, most alkaline proteases produced by genus *Bacillus* are well-known for their industrial potential [14,18] and the results of this newly isolated bacterium agree with this concept.

Production conditions for Mar64P were optimized in submerged mode at pH 9.0 and 45 °C with 0.5% maltose and 1% tyrosine as carbon and energy sources, respectively (Table 1). The best Mar64P productivity was achieved at 60 h after bacterial growth peaked at 36 h which is indicative of a secondary metabolite. This agrees with many studies reporting maximal productivity of these enzymes by the beginning of stationary phase, possibly due to limited nutrients and the poor role played by alkaline proteases in bacterial growth. Comparable observations were reported for proteases produced by *B. circulans* [19], *B. subtilis* [6], *B. gibsonii* 6BS15-4 [14], and *B. nakamurai* PL4 [20].

Mar64P was homogenously purified, and the final specific activity was 8.5-fold with a recovery of 12.4%. SDS-PAGE indicated a prominent band at 28 kDa with similarity with the alkaline protease of *B. horikoshii* [21] while it was different from the alkaline proteases of *Priestia megaterium* gasm32 (30 kDa, [10]), *B. alveayuensis* CAS 5 (33 kDa, [22]), *Salmonella typhimurium* UcB5 (35 kDa, [23]), *B. cereus* (35 kDa, [16]), *B. subtilis* D10 (35 kDa, [24]), and *B. subtilis* D9 (48 kDa, [25,26]).

The activity of Mar64P against the substrate azocasein was maximum at pH 11.0 and was pH-stable for 1 h at 7–12. This is comparable with the commercial alkaline proteases

used as additives to liquid and dry detergents where the optimum value for Subtilisin Novo, Subtilisin Carlsberg, and Savinase is pH 10.5, and they are stable at pH 7.0–12.0 [27].

In addition, the protease was active at high temperatures with best catalysis at 55 °C and thermal stability up to 70 °C for 1 h. In coincidence, the commercial detergent proteases Alcalase, Savinase, and Maxatase have optimal values at 50–60 °C [28]. Moreover, at room temperature Mar64P exerted full reactivity, indicating its suitability for wash performance at both higher and lower temperatures [29]. These pH and thermal attributes make Mar64P a promising candidate in detergent, leather, and skin dehairing formulations [14,30].

Metal ions are known as cofactors for special types of alkaline proteases, they can enhance or suppress the enzyme action. In this study, all tested metals were stimulatory, especially magnesium ions. This indicates the suitability of Mar64P for washing in hard water where many cations such as magnesium and calcium are present. Regarding other investigators, many alkaline proteases from the genus *Bacillus* are stimulated by these metal ions [16]. While the alkaline proteases of *B. gibsonii* 6BS15-4 [14] and *Pyxidicoccus* sp. 252 [5] lost 15–20% of initial activity upon exposure to calcium ions for 1 h making them less suitable as detergent additives in hard water.

The impact of protease inhibitors EDTA and PMSF on enzymatic activity give an idea about the characteristics of active moiety and the required cofactors for enzymatic catalysis [30]. The activity of Mar64P was not influenced by the serine protease inhibitor PMSF while EDTA exponentially suppressed the activity starting from 1 mM to 40 mM concentration for 1 h at 55 °C, tentatively suggesting a metalloprotease-type that requires metals for catalysis. While serine protease-type was reported for *Pyxidicoccus* sp. 252 [5], *B. gibsonii* 6BS15-4 [14], *B. cereus* AUST-7 [16]. The later proteases get oxidized by the bleaching ingredients, therefore, are not recommended as detergent additives [30].

To be used as a detergent additive, organic synthesizer, or similar applications, alkaline proteases should show prolonged shelf-life and storage stability. Mar64P showed prominent storage stability at both high and low temperatures till 4 and 8 weeks, respectively. Furthermore, storage in the existence of 20% organic solvents for 15 days conserved most of the activity, especially with acetone, acetonitrile, and isopropanol. This stability coincides with PyCP from *Pyxidicoccus* sp. 252 [5]. The instability in presence of toluene coincides with the protease of *Bacillus megaterium* AU02 [13]. The enzyme of *Bacillus* sp. was inhibited in the existence of ethanol [31]. The maintenance of protease activity in presence of organic solvents may be attributed to the dehydration and precipitation of enzyme molecules that conserve the conformation and structure of enzyme [32]. It is frequently stated that proteases with organic solvent stability are also tolerant to other denaturants. In addition, these types of alkaline proteases have the potential to be employed as biocatalysts in organic synthesis to turn the equilibrium of the reversible reaction between synthesis and hydrolysis of peptides [12]. Besides, recovering back the organic solvents being employed for precipitation of enzyme make the downstream processing rapid and cost-effective on a large scale in industry [33].

The application of alkaline proteases as bioadditives in laundry detergents to substitute the phosphate additives makes them eco-friendlier and more cost-effective [15]. This requires a high degree of compatibility with the ingredient components of detergents that may suppress their activity and stability [34]. The tested Mar64P showed great stability and compatibility with many powdered detergents after 1 h exposure at both room conditions (25 °C) and high-temperature conditions (55 °C). The best compatibility of enzyme was obtained with Omo™, conserving 97.8% and 85.5% of its initial activity at 25 °C and 55 °C, respectively. In other studies, few proteases were found compatible with detergents, especially the liquid type [15,35].

Furthermore, the destaining capability of Mar64P as an additive to the commercial detergent Omo™ was evaluated against various stains such as blood, egg yolk, chocolate, coffee, tea, and sweat. Mar64P plus detergent showed enhanced performance as compared with the detergent alone. They improve the washing efficiency of fibers and hard surfaces by hydrolyzing the proteinaceous material in stains, enhancing emulsification, increasing

the solubility of stains and foaming characteristics, and minimizing the surface tension and redeposition of hydrolyzed proteinaceous material [36].

The raising requirement for silver in the world especially in the medical fields, makes it very difficult to obtain and manufacture new silver. Therefore, recent attention is paid to the recovery and reuse of X-ray and photographic films. The waste X-ray films participate to ~20% of silver manufacture globally. The traditional physical protocol for the regaining of silver from waste films includes burning of films, however, this results in air pollution and bad smell. The conventional bioremediation protocol also includes chemicals that could react with the film and extract silver atoms which is also non ecofreindly, expensive, and time-consuming protocol. The use of alkaline proteases in the hydrolysis of waste X-ray films is a promising green alternative for the recovery of silver [37]. Mar64P successfully hydrolyzed the gelatin layer within 120 min only. This also allows the reuse of the polyester sheet in the manufacturing of new X-ray films or other applications. The recovery of silver from X-ray films within 120 min was also achieved by the alkaline protease of *Bacillus* sp. [17].

For the dehairing stage in leather industry, animal skins are exposed to many chemicals such as sodium sulfide and lime which leak with effluents into the water bodies leading to environmental pollution and adverse effects on the living organisms [38]. For this, there is a raising attention to applying alkaline proteases not only as green substitutes but also to increase leather quality. In addition, the leather industry requires much water however, the physicochemical characteristics of water differ with source and season [39]. For this Mar64P will be suitable based on its activity and stability at wider temperatures and pH ranges in addition to the suppressing effect of metals found in hard water. It was found that hair removal from the skin of sheep was almost completed by Mar64P after 90 min exposure. However, prolonged exposure adversely affected the tissue and hydrolyzed proteins appeared in the surrounding medium. Alkaline proteases can digest collagen, so hair removal must be controlled to maintain quality and avoid skin damage. The exposure time should be controlled to allow attacking the hair below the epidermis and preserved the quality of the skin on an industrial level; however, enzymatic methods are superior to chemical methods, due to less use of hazardous chemicals. This trend enhances the green recycling processes that are ecofriendly and contributes to the development of animal skin tanning processes with higher productivity than before [38].

The removal of hair by alkaline proteases is due to the hydrolysis of proteins, proteoglycans, and glycoproteins at the base of hair follicles. Dehairing of animal skin has been also described by alkaline proteases secreted by *Vibrio* sp. CA1-1, *Bacillus* sp., and *Pyxidicoccus* sp. 252 [5,39]. The alkaline protease of *Bacillus cereus* removed hair after 12 h and after 15 h, brilliant clear holes with hairs excluded are observed on the surface of the skin [16].

## 5. Conclusions

A newly isolated *Bacillus* sp. strain Mar64 was recovered from soil with an extracellular Mar64P alkaline protease active over a wide range of pHs and temperatures. Besides, it was unsuppressed by the metals of hard water and showed prominent storage stability which collectively suggests a promising candidate for many industrial processes. Such inherent stable proteases are very helpful during various applications because they don't need any modification to be stabilized. When used as an additive to some commercial detergents it showed prominent compatibility and destaining performance. In addition, it was able to hydrolyze waste X-ray films efficiently releasing silver and could allow the reuse of the bedding polyester material. Furthermore, it showed potent dehairing activity of animal skin within a short time. Collectively, the present investigation highlights the possible use of the alkaline protease from the local isolate *Bacillus* sp. Mar64 in many applications.

**Supplementary Materials:** The following supporting information can be downloaded at: https: //www.mdpi.com/article/10.3390/fermentation9070637/s1, Table S1: Sites of protease-producing

bacterial isolates in Saudi Arabia screening; Table S2: Sites of thirty-seven alkaline protease–producing bacteria in Saudi Arabia; Table S3: Estimates of Evolutionary Divergence between Sequences of strain *Bacillus* sp. Mar64 using 16SrDNA gene and reference sequences using the Maximum Composite Likelihood model through MEGA11; Figure S1: Source localities of thirty samples for the protease-producing bacteria (ten sites on the map); Figure S2: The geographic coordinates, latitude, and longitude for sources of the most potent alkaline protease-producing bacterial isolates.

**Author Contributions:** Conceptualization, E.K. and A.H.A.; Methodology, M.A.A., E.K., A.H.A., A.I.A., E.A., J.F.B. and S.A.; Software, J.F.B. and S.A.; Validation, E.K. and A.H.A.; Formal Analysis, E.K., A.H.A., A.I.A. and E.A.; Software, J.F.B. and S.A.; Investigation, M.A.A.; Resources, E.K. and A.H.A.; Data Curation, A.I.A. and E.A.; Writing—Original Draft Preparation, E.K.; Writing—Review & Editing, E.K., A.H.A., A.I.A., E.A., J.F.B. and S.A.; Visualization, E.K. and A.H.A.; Supervision, E.K., A.H.A. and J.F.B.; Project Administration and Funding Acquisition, E.K. All authors have read and agreed to the published version of the manuscript.

**Funding:** The authors extend their appreciation to the Deputyship for Research & Innovation, Ministry of Education in Saudi Arabia for funding this research work through project number IF-2020-028-BASRC at Imam Abdulrahman bin Faisal University (IAU)/Basic and Applied Scientific Research Center (BASRC).

**Institutional Review Board Statement:** Not applicable.

**Data Availability Statement:** All data generated and analyzed during this study are included in this published article.

**Conflicts of Interest:** The authors have no conflict of interest in compliance with the journal guidelines.

**Abbreviations**

BLAST, Basic Local Alignment Search Tool; cfu, colony forming unit; DEAE-Sepharose, di-ethylaminoethyl Sepharose; EDTA, ethylenediamine tetraacetic acid; Mar64P, *Bacillus* sp. Mar64 alkaline protease; LB broth, Luria-Bertani broth; PMSF, phenylmethylsulfonyl fluoride; SDS, sodium dodecyl sulfate; SDS-PAGE, Sodium dodecyl-sulfate polyacrylamide gel electrophoresis; TCA, trichloroacetic acid.

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
