# Peer review of "Isolation, Screening, and Identification of Alkaline Protease-Producing Bacteria and Application of the Most Potent Enzyme from Bacillus sp. Mar64"

_fermentation, doi:10.3390/fermentation9070637_

Round 1

Reviewer 1 Report

The manuscript described the identification of a potent alkaline protease Mar64P, from Bacillus proteolyticus Mar64, which exhibited the maximum activity among all the isolated strains. The production of Mar64P was optimized, and the enzyme was purified to study its biochemical properties, such as the effects of temperature, pH, metal ions, inhibitors, organic solvents, and storage conditions. Interestingly, the detergent compatibility was evaluated to study the effectiveness of cotton fabric destaining. In addition, the dehairing activity and the gelatin hydrolysis activity for the recovery of silver from waste X-ray films were investigated. The industrial applicability of Mar64P was intensively reported. It is recommended for publication after modifications. Below are the suggestions for the improvement of the manuscript.

1. Thirty samples from ten sites can be shown on the map as a figure. An additional table is also required to show the sites of one hundred and thirty isolates and the sites of thirty-seven alkaline protease–producing bacteria.

2. The geographic coordinates, such as latitude and longitude, of sources of four potent alkaline protease-producing bacterial isolates should be shown in Table 1.

3. The description of the most potent isolate, B. proteolyticus Mar64, should be consistent and follow the rule. There was not enough evidence, from phenotypic, physiological, genotypic, genomic, and phylogenetic aspects, to demonstrate the isolated strain should be naming B. proteolyticus. It is suggested to use Bacillus sp. Mar64 in this study. Similarly, B. stercoris Mar67 is suggested to change to Bacillus sp. Mar67.

4. The metabolic fingerprint of Bacillus sp. Mar64 could be shown in a table.

5. The genetic distance should be shown in Figure 1.

6. The concentrations of metal ions, EDTA, and PMSF used in Figure 4a should be described in the legend.

7. Six kinds of different detergent brands were selected for the evaluation of detergent compatibility. However, the details of the six detergents should be described in section 2.8 because each brand has many product versions for various purposes. It’s important to provide information on detergent versions a specific brand uses.

8. The visual inspection of the effectiveness of cotton fabric destaining, the dehairing activity, and the gelatin hydrolysis activity was not convincing because the photo-taking conditions vary, for example, the size of the pictures, the light intensity, and others. It is suggested to use optical instruments, such as a spectrophotometer and image analyzer, to make a scientific graph for Figures 7-9.

9. The author's contributions did not follow the rules of the journal by using the CRediT author statements.

Minor editing of English language required.

Author Response

Dear Ms. Nola,

Thank you for your efforts regarding our Manuscript ID: fermentation-2455282 titled: Isolation, screening, and identification of alkaline protease-producing bacteria and application of the most potent enzyme from Bacillus sp. Mar64.

Reviewer 1

  • The manuscript described the identification of a potent alkaline protease Mar64P, from Bacillus proteolyticus Mar64, which exhibited the maximum activity among all the isolated strains. The production of Mar64P was optimized, and the enzyme was purified to study its biochemical properties, such as the effects of temperature, pH, metal ions, inhibitors, organic solvents, and storage conditions. Interestingly, the detergent compatibility was evaluated to study the effectiveness of cotton fabric destaining. In addition, the dehairing activity and the gelatin hydrolysis activity for the recovery of silver from waste X-ray films were investigated. The industrial applicability of Mar64P was intensively reported. It is recommended for publication after modifications. Below are the suggestions for the improvement of the manuscript.

Response: thanks for this positive opinion.

  1. Thirty samples from ten sites can be shown on the map as a figure. An additional table is also required to show the sites of one hundred and thirty isolates and the sites of thirty-seven alkaline protease–producing bacteria.

Response: Done. See the supplementary material for the geographic localites of soil samples and protease-producing bacteria (Supplementary Figure S1 and S2 in addition to Supplementary Table S1 and S2).

  1. The geographic coordinates, such as latitude and longitude, of sources of four potent alkaline protease-producing bacterial isolates should be shown in Table 1.

Response: Done.

  1. The description of the most potent isolate, proteolyticus Mar64, should be consistent and follow the rule. There was not enough evidence, from phenotypic, physiological, genotypic, genomic, and phylogenetic aspects, to demonstrate the isolated strain should be naming B. proteolyticus. It is suggested to use Bacillus sp. Mar64 in this study. Similarly, B. stercoris Mar67 is suggested to change to Bacillus sp. Mar67.

Response: Done for both species of bacteria.

  1. The metabolic fingerprint of Bacillus sp. Mar64 could be shown in a table.

Response: we wrote these characteristics as a paragraph to minimize the number of tables.

  1. The genetic distance should be shown in Figure 1.

Response: Done, also Estimates of Evolutionary Divergence between Sequences of strain Bacillus sp. Mar64 using 16SrDNA gene and reference sequences using the Maximum Composite Likelihood model through MEGA11 is done (Supplementary Table S3).

  1. The concentrations of metal ions, EDTA, and PMSF used in Figure 4a should be described in the legend.

Response: done.

  1. Six kinds of different detergent brands were selected for the evaluation of detergent compatibility. However, the details of the six detergents should be described in section 2.8 because each brand has many product versions for various purposes. It’s important to provide information on detergent versions a specific brand uses.

Response: done. Persil ProClean®, Dac Multidetergent®, Bonux 3-in-1®, Omo Laundry Detergent®, Tide Simply Clean & Fresh®, and Ariel Matic Front Load Washing®.

  1. The visual inspection of the effectiveness of cotton fabric destaining, the dehairing activity, and the gelatin hydrolysis activity was not convincing because the photo-taking conditions vary, for example, the size of the pictures, the light intensity, and others. It is suggested to use optical instruments, such as a spectrophotometer and image analyzer, to make a scientific graph for Figures 7-9.

Response: we will consider this in the future studies.

  1. The author's contributions did not follow the rules of the journal by using the CRediT author statements.

Response: done.

  • Comments on the Quality of English Language

Minor editing of English language required.

Response: done.

Reviewer 2 Report

The manuscript describes isolation and characterization of alkaline protease from a new source. The properties of the enzyme make it a promissing additive to washing detergents and in other inductrial applications.

The study was well designed, conducted and described. The remarks below are only formal.

1. Show centrifugal force in "g" instead of rpm, as the former does not depend on rotor size.

2. The unit of enzyme activity is not well defined, please consult textbooks for the definition.

3. Please write "metal" ion not "metallic".

4. All numerical values shown should be rounded off to SIGNIFICANT digits.

5. Please write "TENTATIVELY classified as metalloprotease" because the information provided is insufficient for a rigirius classification.

The English is satisfactory.

Author Response

Dear Ms. Nola,

Thank you for your efforts regarding our Manuscript ID: fermentation-2455282 titled: Isolation, screening, and identification of alkaline protease-producing bacteria and application of the most potent enzyme from Bacillus sp. Mar64.

Reviewer 2

  • The manuscript describes isolation and characterization of alkaline protease from a new source. The properties of the enzyme make it a promissing additive to washing detergents and in other inductrial applications.

Response: thanks for this positive opinion.

  • The study was well designed, conducted and described. The remarks below are only formal.

Response: thanks for this.

  1. Show centrifugal force in "g" instead of rpm, as the former does not depend on rotor size.

Response: done for centrifugation while not converted in case of shaking incubators because the last always expressed in rpm.

  1. The unit of enzyme activity is not well defined, please consult textbooks for the definition.

Response: The unit of enzyme activity has many definitions and is different from author to author. For example, some express it as the amount of enzyme liberating a fixed amount of end product per unit time and volume. Others as in our case express the unit of enzyme activity as the amount of enzyme producing product of 0.01 absorbance.

  1. Please write "metal" ion not "metallic".

Response: done

  1. All numerical values shown should be rounded off to SIGNIFICANT digits.

Response: done, unified to the nearest decimal point.

  1. Please write "TENTATIVELY classified as metalloprotease" because the information provided is insufficient for a rigirius classification.

Response: done.

  • Comments on the Quality of English Language

The English is satisfactory.

Response: thanks

Editorial comments:

(I) Please revise your manuscript according to the referees’ comments and

upload the revised file within 5 days.

Response: done

(II) Please use the version of your manuscript found at the above link for

your revisions.

Response: done

(III) Please check that all references are relevant to the contents of the

manuscript.

Response: done

(IV)  Any revisions to the manuscript should be highlighted, such that any

changes can be easily reviewed by editors and reviewers.

Response: done

(V) Please provide a short cover letter detailing your changes for the

editors’ and referees’ approval.

Response: done

Kind regards,

Dr. Essam Kotb
